# Fidaxomicin jams *Mycobacterium tuberculosis* RNA polymerase motions needed for initiation via RbpA contacts

**Hande Boyaci[1†], James Chen[1†], Mirjana Lilic[1], Margaret Palka[2], Rachel Anne Mooney[2], Robert Landick[2,3], Seth A Darst[1]\*, Elizabeth A Campbell[1]\***

[1]The Rockefeller University, New York, United States; [2]Department of Biochemistry, University of Wisconsin-Madison, Madison, United States; [3]Department of Bacteriology, University of Wisconsin-Madison, Madison, United States

**Abstract** Fidaxomicin (Fdx) is an antimicrobial RNA polymerase (RNAP) inhibitor highly effective against *Mycobacterium tuberculosis* RNAP in vitro, but clinical use of Fdx is limited to treating *Clostridium difficile* intestinal infections due to poor absorption. To identify the structural determinants of Fdx binding to RNAP, we determined the 3.4 Å cryo-electron microscopy structure of a complete *M. tuberculosis* RNAP holoenzyme in complex with Fdx. We find that the actinobacteria general transcription factor RbpA contacts fidaxomycin, explaining its strong effect on *M. tuberculosis*. Additional structures define conformational states of *M. tuberculosis* RNAP between the free apo-holoenzyme and the promoter-engaged open complex ready for transcription. The results establish that Fdx acts like a doorstop to jam the enzyme in an open state, preventing the motions necessary to secure promoter DNA in the active site. Our results provide a structural platform to guide development of anti-tuberculosis antimicrobials based on the Fdx binding pocket.
DOI: https://doi.org/10.7554/eLife.34823.001

**\*For correspondence:**
darst@rockefeller.edu (SAD);
campbee@rockefeller.edu (EAC)

[†]These authors contributed equally to this work

**Competing interests:** The authors declare that no competing interests exist.

## Introduction

The bacterial RNA polymerase (RNAP) is a proven target for antibiotics. The rifamycin (Rif) class of antibiotics, which inhibit RNAP function, is a lynchpin of modern tuberculosis (TB) treatment (*Chakraborty and Rhee, 2015*). TB, caused by the infectious agent *Mycobacterium tuberculosis* (*Mtb*), is responsible for almost 2 million deaths a year. It is estimated that one third of the world is infected. Mortality from TB is increasing, partly due to the emergence of strains resistant to Rifs (Rif[R]) (*Zumla et al., 2015*). Hence, additional antibiotics against Rif[R] *Mtb* are needed.

Fidaxomicin (Fdx; also known as Dificimicin, lipiarmycin, OPT-80, PAR-101, or tiacumicin), an antimicrobial in clinical use against *Clostridium difficile* (*Cdf*) infection (*Venugopal and Johnson, 2012*), functions by inhibiting the bacterial RNAP (*Talpaert et al., 1975*). Fdx targets the RNAP 'switch region', a determinant for RNAP inhibition that is distinct from the Rif binding pocket (*Srivastava et al., 2011*), and Fdx does not exhibit cross-resistance with Rif (*Gualtieri et al., 2009, 2006*; *Kurabachew et al., 2008*; *O'Neill et al., 2000*). The switch region sits at the base of the mobile RNAP clamp domain and, like a hinge, controls motions of the clamp crucial for DNA loading into the RNAP active-site cleft and maintaining the melted DNA in the channel (*Chakraborty et al., 2012*; *Feklistov et al., 2017*). Fdx is a narrow spectrum antibiotic that inhibits Gram-positive anae-robes and mycobacteria (including *Mtb*) much more potently than Gram-negative bacteria (*Kurabachew et al., 2008*; *Srivastava et al., 2011*), but the clinical use of Fdx is limited to intestinal infections due to poor bioavailability (*Venugopal and Johnson, 2012*). Addressing this limitation requires understanding the structural and mechanistic basis for Fdx inhibition, which is heretofore

**eLife digest** Tuberculosis (TB) is an infectious disease that affects over ten million people every year. The *Mycobacterium tuberculosis* bacteria that cause the disease spread through the air from one person to another and mainly infect the lungs. Although curable, TB is difficult to eradicate because it is remarkably widespread, with one third of the world's population estimated to carry the bacteria.

Treatment for TB involves a mix of antibiotics that should be taken for several months to a year. The number of multidrug-resistant TB cases, where the infection is not treatable by the common cocktail of antibiotics, is rapidly increasing. There is therefore a need to discover new drugs that can kill the *M. tuberculosis* bacteria.

An antibiotic called fidaxomicin is used to treat intestinal infections. Although it can kill *Mycobacterium tuberculosis* cells in culture, it is not absorbed from the intestines to the blood and thus cannot reach the lungs to kill the bacteria. It may be possible to change the structure of the drug so that it can enter the bloodstream. Before this can be done, researchers need to understand exactly how fidaxomicin kills the bacteria so that they know which parts of the drug they can alter without making it less effective.

Fidaxomicin kills bacterial cells by binding to an enzyme called RNA polymerase. The antibiotic prevents the enzyme from reading and 'transcribing' DNA to form molecules that are essential for life. To learn more about how fidaxomicin has this effect, Boyaci, Chen et al. used cryo-electron microscopy to look at structures of the *M. tuberculosis* RNA polymerase in different states, including when it was bound to fidaxomicin.

The structures reveal the chemical details of the interactions between the RNA polymerase and the antibiotic. The two molecules bind to each other through a region of the RNA polymerase that is unique to *M. tuberculosis* and closely related bacteria. Fidaxomicin acts like a doorstop to jam the RNA polymerase in an open state that cannot bind to DNA and transcribe genes.

Medicinal chemists could now build on these findings to develop new drugs that might treat TB, either by modifying fidaxomicin or designing new antibiotics that bind to the same region of the RNA polymerase. Because the fidaxomicin-binding region of the RNA polymerase is specific to *M. tuberculosis* new antibiotics could be tailored towards the bacteria that have a minimal effect on a patient's normal gut bacteria.

DOI: https://doi.org/10.7554/eLife.34823.002

unknown. Here, we used single-particle cryo-electron microscopy (cryo-EM) to determine structures of *Mtb* transcription initiation complexes in three distinct conformational states, including a complex with Fdx at an overall resolution of 3.4 Å. The results define the molecular interactions of *Mtb* RNAP with Fdx as well as the mechanistic basis of inhibition, and establish that RbpA, an Actinobacteria-specific general transcription factor (GTF), is crucial to the sensitivity of *Mtb* to Fdx.

## Results

### Fdx potently inhibits mycobacterial TICs in vitro

Fdx has potent inhibitory activity against multi-drug-resistant *Mtb* cells and the in vivo target is the RNAP (*Kurabachew et al., 2008*). To our knowledge, the in vitro activity of Fdx against mycobacterial RNAPs has not been reported. RbpA, essential in *Mtb*, is a component of transcription initiation complexes (TICs) that tightly binds the primary promoter specificity $\sigma^A$ subunit of the RNAP holoenzyme (holo) (*Bortoluzzi et al., 2013*; *Forti et al., 2011*; *Hubin et al., 2017a*, *2015*; *Tabib-Salazar et al., 2013*). We therefore compared Fdx inhibition of mycobacterial RNAPs containing core RNAP combined with $\sigma^A$ ($\sigma^A$-holo) and RbpA with inhibition of *Escherichia coli* (*Eco*) $\sigma^{70}$-holo using a quantitative abortive initiation assay (*Davis et al., 2015*). Fdx inhibited *Mtb* and *M. smegmatis* (*Msm*) transcription at sub-µM concentrations, whereas inhibition of an *Mtb* TIC containing Fdx-resistant (Fdx$^R$) RNAP ($\beta^{Q1054H}$) (*Kurabachew et al., 2008*) required a nearly two orders of magnitude higher concentration of Fdx. *Eco* RNAP was inhibited even less effectively by another order of magnitude (*Figure 1A*, *Figure 1—figure supplement 1A*).

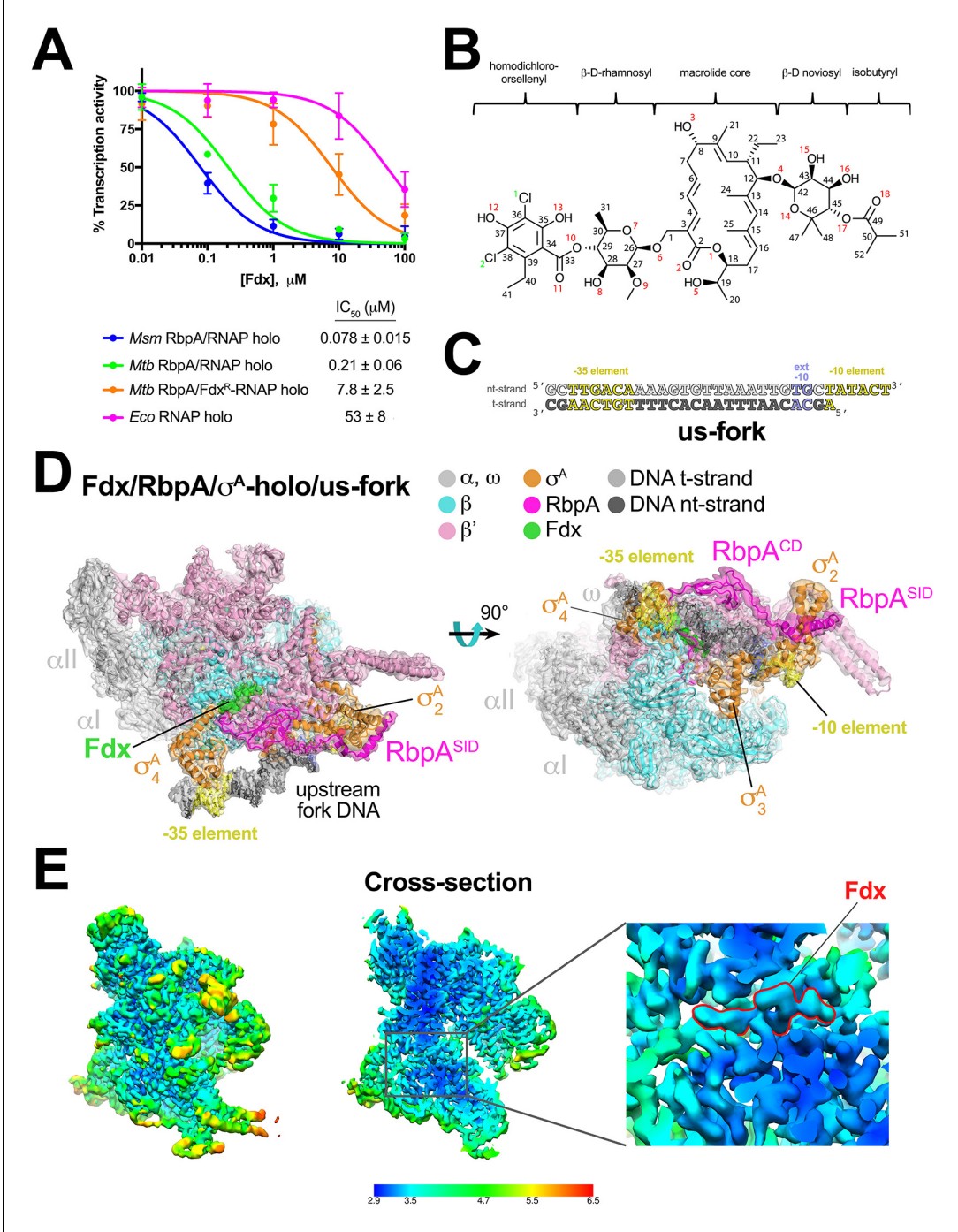

**Figure 1.** Structure of an *Mtb* RbpA/TIC with Fdx at 3.4 Å resolution. (**A**) Fdx inhibits mycobacterial RbpA/σ^A-holo transcription greater than 250-fold more effectively than *Eco*σ^70-holo in in vitro abortive initiation assays. The error bars denote the standard error from a minimum of three experiments (for some points, the error bars are smaller than the width of the point and are not shown). (**B**) Chemical structure of Fdx (*Serra et al., 2017*). (**C**) Synthetic us-fork promoter fragment used for cryo-EM experiments. The DNA sequence is derived from the full con promoter (*Gaal et al., 2001*). The nontemplate-strand DNA (top strand) is colored light gray; the template-strand DNA (bottom strand), dark grey. The −35 and −10 elements are shaded yellow. The extended −10 (*Keilty and Rosenberg, 1987*) is colored violet. (**D**) The 3.4 Å resolution cryo-EM density map of the Fdx/RbpA/σ^A-holo/us-fork complex is rendered as a transparent surface colored as labeled. Superimposed is the final refined model; proteins are shown as a backbone ribbon, Fdx and the nucleic acids are shown in stick format. (**E**) Views of the cryo-EM map colored by local resolution based on blocres calculation (*Cardone et al., 2013*). The left view shows

*Figure 1 continued on next page*

*Figure 1 continued*

the entire map, while the middle view shows a cross-section of the map sliced at the level of the Fdx binding pocket. The boxed region is magnified on the right. Density for the Fdx molecule is outlined in red.

DOI: https://doi.org/10.7554/eLife.34823.003

The following figure supplements are available for figure 1:

**Figure supplement 1.** Abortive Transcription assays to determine $IC_{50}$ for Fdx.

DOI: https://doi.org/10.7554/eLife.34823.004

**Figure supplement 2.** Data processing pipeline for the cryo-EM movies of the Fdx/RbpA/$\sigma^A$-holo/us-fork complexes.

DOI: https://doi.org/10.7554/eLife.34823.005

**Figure supplement 3.** Fdx/RbpA/$\sigma^A$-holo/us-fork class.

DOI: https://doi.org/10.7554/eLife.34823.006

## Cryo-EM structure of the Fdx/RbpA/$\sigma^A$-holo complex

We used single-particle cryo-EM to examine the complex of *Mtb* RbpA/$\sigma^A$-holo with and without Fdx (*Figure 1B*). Preliminary analyses revealed that the particles were prone to oligomerization, which was reduced upon addition of an upstream-fork (us-fork) junction promoter DNA fragment (*Figure 1C*). We sorted nearly 600,000 cryo-EM images of individual particles into two distinct classes, each arising from approximately half of the particles (*Figure 1—figure supplement 2*).

The first class comprised *Mtb* RbpA/$\sigma^A$-holo with one us-fork promoter fragment and bound to Fdx. The cryo-EM density map was computed to a nominal resolution of 3.4 Å (*Figure 1D*, *Figure 1—figure supplement 3*, *Supplementary file 1*). The us-fork promoter fragment was bound outside the RNAP active site cleft, as expected, with the −35 and −10 promoter elements engaged with the $\sigma^A_4$ and $\sigma^A_2$ domains, respectively (*Figure 1D*). Local resolution calculations (*Cardone et al., 2013*) indicated that the central core of the structure, including the Fdx binding determinant and the bound Fdx, was determined to 2.9–3.4 Å resolution (*Figure 1E*).

## Cryo-EM structure of a *Mtb* RPo mimic

The second class comprised *Mtb* RbpA/$\sigma^A$-holo bound to two us-fork promoter fragments but without Fdx to a nominal resolution of 3.3 Å (*Figure 2A*, *Figure 2—figure supplement 1*, *Supplementary file 1*). One us-fork promoter fragment bound upstream from the RNAP active site cleft as in the previous class, but a second us-fork promoter fragment bound the RNAP downstream duplex DNA binding channel, with the 5-nucleotide 3'-overhang (*Figure 1C*) engaged with the RNAP active site (as the template strand) like previously characterized 3'-tailed templates (*Gnatt et al., 2001*; *Kadesch and Chamberlin, 1982*). Local resolution calculations (*Cardone et al., 2013*) indicated that the central core of the structure was determined to between 2.8–3.2 Å resolution (*Figure 2B*). The overall conformation of this protein complex and its engagement with the upstream and downstream DNA fragments was very similar to the crystal structure of a full *Msm* open promoter complex (RPo) (*Hubin et al., 2017b*) with one exception (see below). We will therefore call this complex an *Mtb* RbpA/RPo mimic.

## The RbpA N-terminal tail invades the RNAP active site cleft

RbpA comprises four structural elements, the N-terminal tail (NTT), the core domain (CD), the basic linker, and the sigma interacting domain (SID) (*Bortoluzzi et al., 2013*; *Hubin et al., 2017a*; *Tabib-Salazar et al., 2013*). Our previous crystal structures of *Msm* TICs containing RbpA showed that the RbpA$^{SID}$ interacts with the $\sigma^A_2$ domain, the RbpA$^{BL}$ establishes contacts with the promoter DNA phosphate backbone just upstream of the −10 element, and the RbpA$^{CD}$ interacts with the RNAP β' Zinc-Binding-Domain (ZBD) (*Hubin et al., 2017a*, *2017b*). Density for the RbpA$^{NTT}$ (RbpA residues 1–25) was never observed in the crystal structures and was presumed to be disordered. In striking contrast to the crystal structures, both cryo-EM structures reveal density for the RbpA$^{NTT}$, which unexpectedly threads into the RNAP active site cleft between the ZBD and $\sigma^A_4$ domains and snakes through a narrow channel towards the RNAP active site $Mg^{2+}$ (*Figure 3*). On its path, conserved residues of the RbpA$^{NTT}$ interact with conserved residues of the σ-finger ($\sigma_{3.2}$-linker) on one wall of the channel, and with conserved residues of the ZBD and β'lid on the other wall (*Figure 3C*).

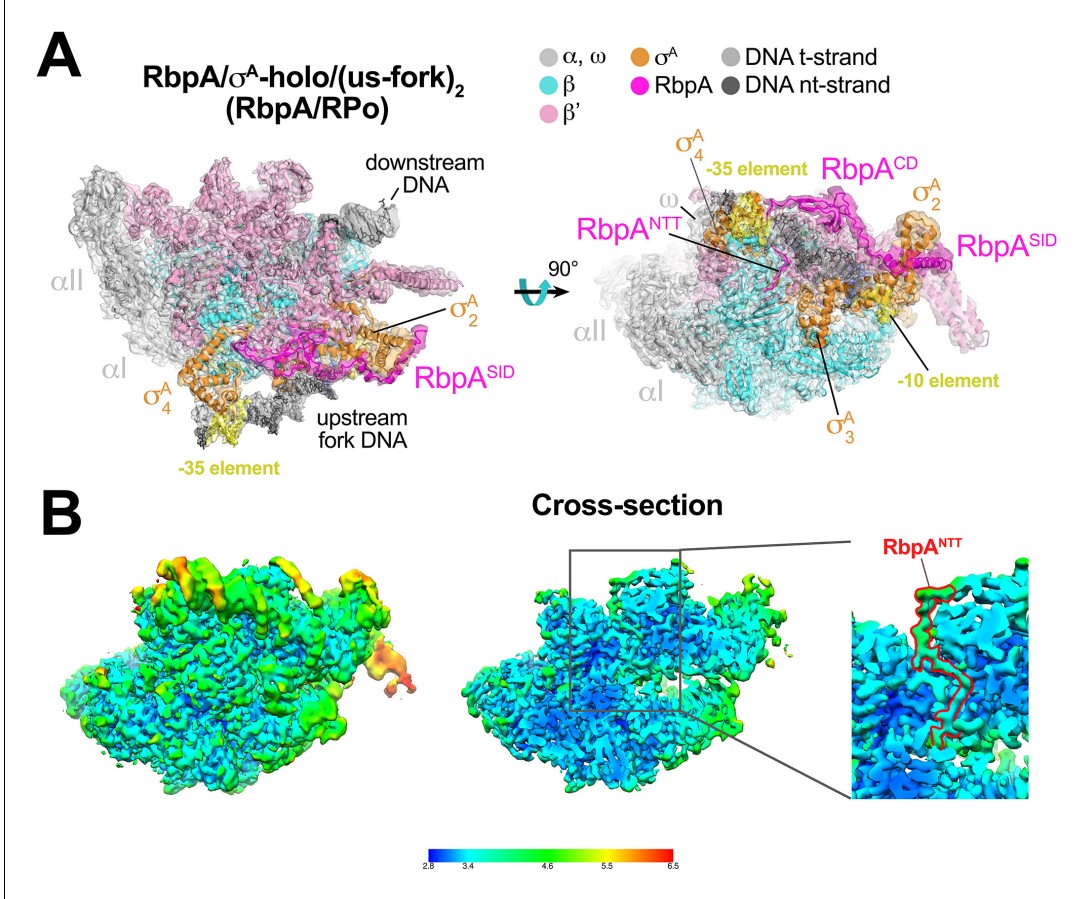

**Figure 2.** Structure of an *Mtb* RbpA/RPo mimic at 3.3 Å resolution. (**A**) The 3.3 Å resolution cryo-EM density map of the RbpA/σ$^A$-holo/(us-fork)$_2$ complex (RbpA/RPo mimic) is rendered as a transparent surface colored as labeled. Superimposed is the final refined model; proteins are shown as a backbone ribbon, nucleic acids are shown in stick format. (**B**) Views of the *Mtb* RbpA/RPo mimic cryo-EM map colored by local resolution based on blocres calculation (*Cardone et al., 2013*). The left view shows the entire map, while the middle view shows a cross-section of the map sliced at the level of the RbpA$^{NTT}$. The boxed region is magnified on the right. Density for the RbpA$^{NTT}$ is outlined in red.

DOI: https://doi.org/10.7554/eLife.34823.007

The following figure supplement is available for figure 2:

**Figure supplement 1.** RbpA/σ$^A$-holo/(us-fork)$_2$ class.

DOI: https://doi.org/10.7554/eLife.34823.008

The most N-terminal RbpA residues visible in the cryo-EM structures (A2 in the Fdx complex, R4 in the RPo) sit near the tip of the σ-finger where it makes its closest approach to the RNAP active site, too far (25 Å) to play a direct role in RNAP catalytic activity or substrate binding. The σ-finger plays an indirect role in transcription initiation, stimulating de novo phosphodiester bond formation by helping to position the t-strand DNA (*Kulbachinskiy and Mustaev, 2006*; *Zhang et al., 2012*). The σ-finger is also a major determinant of abortive initiation, playing a direct role in initiation and promoter escape by physically blocking the path of the elongating RNA transcript before σ release (*Cashel et al., 2003*; *Murakami et al., 2002*). The intimate association of the RbpA$^{NTT}$ with the σ-finger (*Figure 3C*) suggests that the RbpA$^{NTT}$ also plays a role in these processes of *Mtb* RNAP initiation. This is consistent with our findings that the RbpA$^{NTT}$ does not strongly affect RPo formation but plays a significant role in in vivo gene expression in *Msm* (*Hubin et al., 2017a*). This location of the RbpA$^{NTT}$ explains the high Fdx sensitivity of *Mtb* RNAP (see below).

## Fdx interacts with RNAP, σ$^A$, and RbpA

The reconstruction from the Fdx-bound class (*Figure 1D*) reveals unambiguous density for Fdx (*Figure 4A*) and defines Fdx-interacting residues from four protein components of the complex, β,

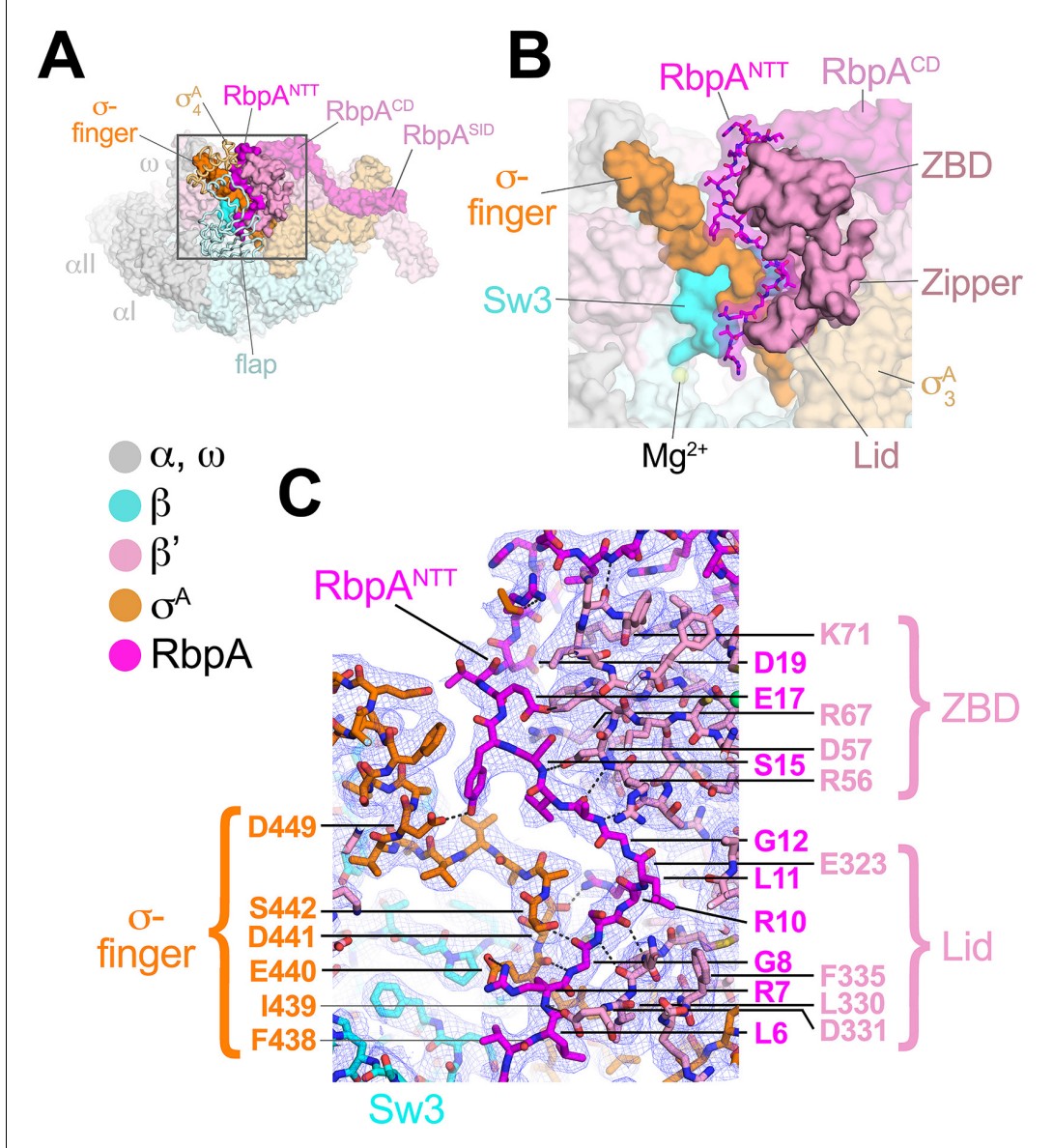

**Figure 3.** The RbpA$^{NTT}$ interacts with conserved structural elements in the RNAP active site cleft. (**A**) An overview of the RbpA/RPo structure is shown as a color-coded molecular surface (color-coding denoted in the key) except the β flap and σ$^A_4$ domain are shown as backbone worms, revealing the RbpA$^{NTT}$ (magenta) underneath. The DNA fragments are not shown. The boxed region is magnified in panel (**B**). (**B**) Magnified view of the boxed region from panel (**A**). The RbpA$^{NTT}$ is shown in stick format with a transparent molecular surface. Conserved RNAP structural elements that interact with the RbpA$^{NTT}$ are highlighted (βSw3, β'ZBD, β'Zipper, β'Lid, and σ-finger). (**C**) Further magnified view showing the cryo-EM density (blue mesh) with the superimposed model. Conserved residues of the RbpA$^{NTT}$ are labeled, along with conserved residues of the β'ZBD, β'Lid, and σ-finger that interact with the RbpA$^{NTT}$.

DOI: https://doi.org/10.7554/eLife.34823.009

β', σ$^A$, and RbpA, including six water molecules, four of which mediate Fdx/RNAP interactions (*Figure 4A,B*). Fdx binding to the TIC buries a large accessible surface area of 4,800 Å$^2$ (β, 2,100 Å$^2$; β', 2,000 Å$^2$; σ$^A$, 300 Å$^2$; RbpA, 330 Å$^2$). Fdx forms direct hydrogen bonds with nine residues (βQ1054, βD1094, βT1096, βK1101, β'R84, β'K86, β'R89, β'E323, and β'R412) and water-mediated interactions with four (β'R89, β'D404, β'Q415, and RbpA-E17). Notably, the Fdx/RNAP interaction is stabilized by two cation-π interactions between β'R84 and the aromatic ring of the Fdx

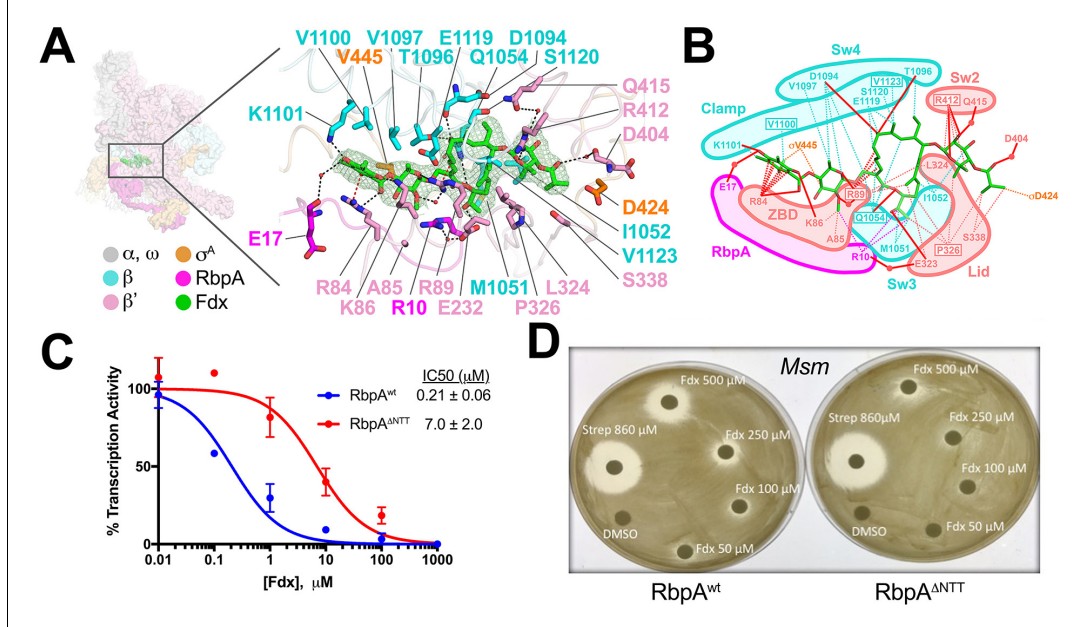

**Figure 4.** Structural basis for Fdx inhibition of *Mtb* transcription and the role of the RbpA^NTT. (A) (left) Overview of the Fdx/RbpA/σ^A-holo/us-fork structure, shown as a molecular surface (the DNA is not shown). The boxed region is magnified on the right. (right) Magnified view of the Fdx binding pocket at the same orientation as the boxed region on the left. Proteins are shown as α-carbon backbone worms. Residues that interact with Fdx are shown in stick format. Fdx is shown in stick format with green carbon atoms. Water molecules are shown as small pink spheres. Hydrogen-bonds are indicated by dashed gray lines. Cation-π interactions (between β'R84 and the aromatic ring of the Fdx homodichloroorsellinic acid moiety and β'R89 and the conjugated double-bond system centered between C4 and C5 of the macrolide core) are represented by red dashed lines. (B) Schematic summary of the Fdx contacts with σ^A-holo and RbpA. Fdx is shown in stick format with green carbon atoms. Thin dashed lines represent van der Waals contacts (≤4.5 Å), thick red lines represent hydrogen bonds (<4 Å). The thin dashed red lines denote cation-π interactions. (C) The RbpA^NTT is required for optimal inhibition of *Mtb* transcription by Fdx in in vitro abortive initiation assays. The error bars denote the standard error from a minimum of three experiments (for some points, the error bars are smaller than the width of the point and are not shown). (D) Zone of inhibition assays with *Msm* cells show that loss of the RbpA-NTT (RbpA^ΔNTT) leads to loss of Fdx sensitivity in vivo.

DOI: https://doi.org/10.7554/eLife.34823.010

homodichloroorsellinic acid moiety (*Figure 1B*) and β'R89 and the conjugated double-bond system centered between C4 and C5 of the macrolide core (*Figures 1B* and *4A,B*). Fdx interacts with residues from eight distinct structural elements (*Lane and Darst, 2010*) of the initiation complex (βSw3, βSw4, β residues belonging to the clamp, β'ZBD, β'lid, β'Sw2, the σ-finger, and the Rbpa^NTT (*Figure 4A,B*).

Amino-acid substitutions conferring Fdx^R have been identified in RNAP β or β' subunits from *Bacillus subtilis* (*Gualtieri et al., 2006*), *Cdf* (*Kuehne et al., 2017*), *Enterococcus faecalis* (*Gualtieri et al., 2009*), and *Mtb* (*Kurabachew et al., 2008*), corresponding to *Mtb* RNAP β residues Q1054 (Sw3), V1100 and V1123 (clamp), and β' residues R89 (ZBD), P326 (lid), and R412 (Sw2). The structure shows that each of these residues makes direct interactions with Fdx (*Figure 4A,B*). All five chemical moieties of Fdx (*Figure 1B*) interact with at least one RNAP residue that confers Fdx^R when mutated (*Figure 4B*), suggesting that each moiety may be important for Fdx action.

## The RbpA^NTT is critical for Fdx potency against mycobacterial RNAP in vitro and in vivo

In addition to the β and β' subunits, Fdx interacts with residues of the σ-finger (D424 and V445; *Figure 4A,B*). Finally and unexpectedly, Fdx contacts residues from the RbpA^NTT (*Figure 4A,B*). To test the functional importance of the RpbA^NTT for Fdx inhibition in vitro, we compared Fdx inhibition of *Mtb*σ^A-holo with either RbpA or RbpA with the NTT truncated (RbpA^ΔNTT) in the abortive initiation assay (*Figure 1—figure supplement 1B*). Truncation of the RbpA-NTT caused a 35-fold increase in resistance to Fdx (*Figure 4C*).

RbpA is essential in *Mtb* and *Msm*, but strains carrying RbpA^ΔNTT are viable (*Hubin et al., 2017a*), allowing us to test the role of the RbpA^NTT in Fdx growth inhibition of *Msm* cells. We performed zone of inhibition assays on two *Msm* strains that are isogenic except one harbors wild-type RbpA (RbpA^wt) and the other RbpA^ΔNTT (*Hubin et al., 2017a*). The *Msm* RbpA^ΔNTT strain grew considerably slower on plates, taking approximately twice the time as the wild-type *Msm* to reach confluency. Despite the growth defect, the RbpA^ΔNTT strain was significantly less sensitive to Fdx (*Figure 4D*). Discs soaked with up to 250 μM Fdx did not produce inhibition zones with RbpA^ΔNTT but inhibition zones were apparent with RbpA^wt. At 500 μM Fdx, the inhibition zone for RbpA^ΔNTT was significantly smaller than for RbpA^wt. By contrast, 860 μM streptomycin, a protein synthesis inhibitor, produced equal inhibition zones for the RbpA^wt and RbpA^ΔNTT strains. We conclude that the essential role of RbpA in *Mtb* transcription is key to the relatively high sensitivity of *Mtb* cells to Fdx.

## Fdx traps an open-clamp conformation

The RNAP switch regions are thought to act as hinges connecting the mobile clamp domain to the rest of the RNAP (*Gnatt et al., 2001*; *Lane and Darst, 2010*). Bacterial RNAP inhibitors myxopyronin, corallopyronin, and ripostatin bind Sw1 and Sw2 and stabilize a closed-clamp conformation of the RNAP (*Belogurov et al., 2009*; *Mukhopadhyay et al., 2008*). The Fdx binding determinant does not overlap the sites for these other inhibitors, but the Fdx interactions with the Sw2, Sw3, and Sw4 regions (*Figure 4A,B*) suggest that Fdx may influence the clamp conformation as well.

To understand the role of Fdx in clamp movement without the complication of DNA binding in the RNAP active site cleft, we determined cryo-EM structures of *Mtb* RbpA/σ^A-holo without DNA, with Fdx and without Fdx. Although the particles in the original cryo-EM datasets of *Mtb* RbpA/σ^A-holo were prone to oligomerization, we used 2D classification to isolate single particles and determined reconstructions of *Mtb* RbpA/σ^A-holo without DNA and with Fdx (overall 6.5 Å resolution from 21,000 particles, *Figure 5A*, *Figure 5—figure supplement 1A–E*) and without Fdx (overall 5.2 Å resolution from 88,000 particles; *Figure 5A*, *Figure 5—figure supplement 1F–I*). The cryo-EM density maps were of sufficient detail to visualize the bound antibiotic in the Fdx complex (*Figure 5—figure supplement 1E*) and to determine the domain organization (including the clamp conformation) by rigid-body refinement (*Figure 5A*). Thus, we were able to compare the RNAP conformational states from four solution complexes of the same RNAP in the absence of crystal packing forces (*Figure 5B*).

The four structures were superimposed by the structural core module (*Supplementary file 2*), comprising the ω subunit and highly conserved β and β′ regions in or near the active center that have not been observed to undergo significant conformational changes in dozens of RNAP structures. Using the RPo structure (*Figure 2A*) as a reference, the structures superimposed with rmsds < 0.4 Å over at least 898 aligned α-carbon (Cα) atoms of the structural core module, but rmsds > 9 Å for 461 Cα-atoms of the clamp modules (*Supplementary file 2*), indicating large shifts of the clamp module with respect to the rest of the RNAP in the different complexes.

Alignment of the structures revealed that the clamp conformational changes could be characterized as rigid body rotations about a common rotation axis (*Figure 5B*). Assigning a clamp rotation angle of 0° (closed clamp) to the RPo structure (blue, *Figure 5B*), the RbpA/σ^A-holo clamp is rotated open by about 12° (green, *Figure 5B*). Because this complex is not interacting with any other ligands that might be expected to alter the clamp conformation (such as Fdx or DNA), we will refer to this as the 'relaxed' clamp conformation. The two Fdx-bound complexes, with or without us-fork DNA, show further opening of the clamp (14° and 15°, respectively; orange and red in *Figure 5B*).

## Fdx acts like a doorstop to stabilize the open-clamp conformation

In the high-resolution Fdx/TIC structure (*Figure 1D*), Fdx binds in a narrow gap between the open clamp module and the rest of the RNAP (*Figure 5C*). Examination of the high-resolution RPo (closed clamp) structure reveals that clamp closure pinches off the Fdx binding pocket (*Figure 5D*) - Fdx can only bind to the open-clamp conformation of RNAP. We thus conclude that Fdx acts like a doorstop, binding and stabilizing the open-clamp conformation.

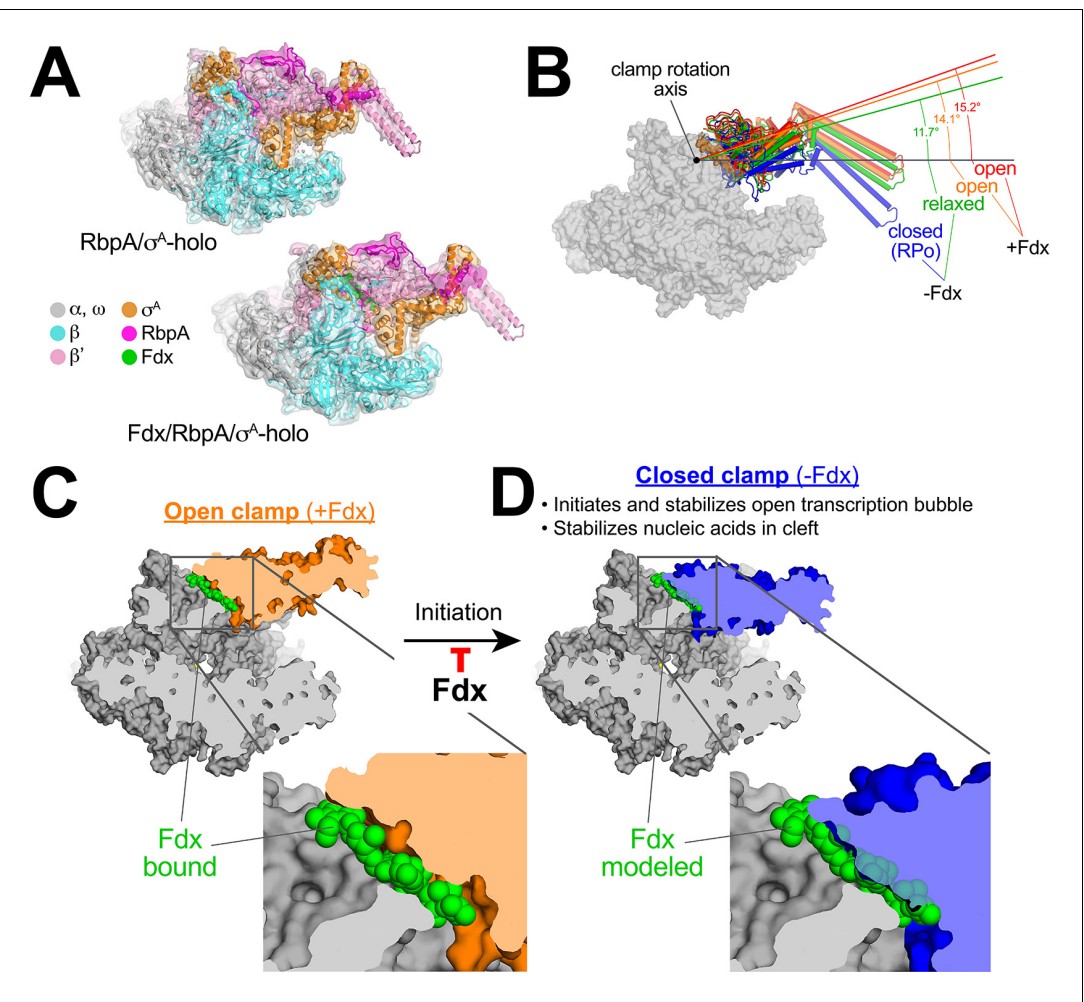

**Figure 5.** Mechanism of Fdx inhibition of bacterial RNAP. (**A**) Cryo-EM density maps and superimposed refined models for *Mtb* RbpA/σ^A-holo (5.2 Å resolution) and *Mtb* Fdx/RbpA/σ^A-holo (6.5 Å resolution). (**B**) RNAP clamp conformational changes for four cryo-EM structures determined in this work. The RbpA/RPo (*Figure 2A*) structure was used as a reference to superimpose the other structures via α-carbon atoms of the structural core module (*Supplementary file 2*), revealing a common core RNAP structure (shown as a gray molecular surface) but with large shifts in the clamp modules. The clamp modules are shown as backbone cartoons with cylindrical helices and color-coded (blue, closed clamp of RPo; green, relaxed clamp of RbpA/σ^Aholo; orange, open clamp of Fdx/RbpA/σ^A-holo/us-fork; red, open clamp of Fdx/RbpA/σ^A-holo). The clamp conformational changes can be characterized as rigid body rotations about a rotation axis perpendicular to the page (denoted). The angles of clamp opening for the different structures are shown (relative to the blue closed RPo clamp, 0° opening). (**C**) The core RNAP from the 3.4 Å resolution Fdx/RbpA/σ^A-holo/us-fork structure is shown as a gray molecular surface but with the open clamp colored orange. The structure is sliced at the level of the Fdx binding pocket (the bound Fdx is shown in green). The boxed region is magnified below, showing the tight fit of the Fdx molecule in a narrow gap between the clamp and the rest of the RNAP. (**D**) The core RNAP from the 3.3 Å resolution RbpA/RPo structure is shown as a gray molecular surface but with the closed clamp colored blue. The structure is sliced at the level of the (empty) Fdx binding pocket. Fdx, modeled from the structure shown in (**C**), is shown in green. The boxed region is magnified below. Fdx cannot bind to RNAP with a closed clamp because clamp closure pinches off the Fdx binding site. Clamp closure is required for initiation and stabilization of the transcription bubble (*Feklistov et al., 2017*) and also for stable binding of nucleic acids in the RNAP cleft.

DOI: https://doi.org/10.7554/eLife.34823.011

The following figure supplement is available for figure 5:

**Figure supplement 1.** Cryo-EM of the *Mtb* RbpA/σ^A-holo and Fdx/RbpA/σ^A-holo complexes.
DOI: https://doi.org/10.7554/eLife.34823.012

## Discussion

### Fdx inhibits RNAP by trapping an open-clamp conformation

Clamp dynamics play multiple important roles in the transcription cycle. Motions of the clamp module and the role of the switch regions as hinges were first noted by comparing crystal structures of free RNAPs (*Cramer et al., 2001*; *Zhang et al., 1999*) with the crystal structure of an elongation complex containing template DNA and RNA transcript (*Gnatt et al., 2001*). Binding of the downstream duplex DNA and RNA/DNA hybrid in the RNAP active-site cleft was proposed to close the clamp around the nucleic acids, explaining the high processivity of the transcription elongation complex. Numerous subsequent crystal structures have supported the idea that stable, transcription-competent complexes of RNAP with nucleic acids, either RPo (*Bae et al., 2015*; *Hubin et al., 2017b*; *Zuo and Steitz, 2015*) or elongation complexes (*Gnatt et al., 2001*; *Kettenberger et al., 2004*; *Vassylyev et al., 2007*), correlate with the closed-clamp conformation. Effects of crystal packing forces on clamp conformation, however, cannot always be ruled out. Observations of clamp positions by solution FRET (*Chakraborty et al., 2012*), and more recently in cryo-EM structures (*Bernecky et al., 2016*; *Hoffmann et al., 2015*; *Kang et al., 2017*; *Neyer et al., 2016*) (in the absence of crystal packing forces) have confirmed the relationship between clamp closure and stable nucleic-acid complexes. Clamp motions have also been shown to play a critical role in the process of promoter melting to form the transcription bubble during RPo formation (*Feklistov et al., 2017*). Thus, the trapping of an open-clamp RNAP conformation by Fdx in unrestrained cryo-EM conditions (*Figure 5C*) suggests that Fdx inhibits transcription initiation by preventing clamp motions required for RPo formation, or by not allowing RNAP to form stable transcription-competent complexes with nucleic acids, or both (*Figure 5C,D*). These results are broadly consistent with mechanistic analyses of (*Tupin et al., 2010*) and (*Morichaud et al., 2016*) showing that Fdx blocks promoter melting at an early step but providing RNAP a pre-melted template overcomes the block. These authors proposed that Fdx likely prevented the clamp from closing, again consistent with our structural findings.

### Summary

Our results establish the molecular details of Fdx interactions with the bacterial RNAP (*Figure 4A,B*) and a mechanism of action for Fdx (*Figure 5C,D*). Crucially, the essential actinobacterial GTF RbpA is responsible for the high sensitivity of mycobacterial RNAP to Fdx both in vitro (*Figure 4C*) and in vivo (*Figure 4D*). This new knowledge provides a structural platform for the development of antimicrobials that target the Fdx binding determinant and underscores the need to define structure-activity relationships of drug leads using near-native states, in this case using cryo-EM with the RbpA/$\sigma^A$-holo complex to guide development of effective *Mtb* treatments.

## Materials and methods

### Key resources table

| Reagent type (species) or resource | Designation | Source or reference | Identifiers | Additional information |
|---|---|---|---|---|
| Strain, strain background (*Escherichia coli*) | *Eco* BL21(DE3) | EMD-Millipore (Novagen; Darmstadt, Germany) | | |
| Strain, strain background (*Mycobacterium smegmatis* mc2155) | MGM6029: *Msm* mc2155 rpoC:rpoC-ppx-10his hyg | PMID: 28067618 | | |
| Strain, strain background (*Mycobacterium smegmatis* mc2155) | MGM6234: *Msm* mc2155 ΔrbpA attB:: rbpA(28-114) kan | PMID: 28067618 | | |

*Continued on next page*

*Continued*

| Reagent type (species) or resource | Designation | Source or reference | Identifiers | Additional information |
|---|---|---|---|---|
| Recombinant DNA reagent | pAC22 | PMID: 24713321 | | pET28a derivative. Encodes M. bovis RNAP. β contains a S450Y substitution (RifR) and a short N-terminal insertion at codon 2 (LEGCIL); β′ has C-terminal His8 tag; β and β′ are fused with a short linker (LARHGGSGA) |
| Recombinant DNA reagent | pACYCDuet-1_Ec_rpoZ | PMID: 23389035 | | |
| Recombinant DNA reagent | pET21a-*Eco*σ70 | PMID: 24218560 | | |
| Recombinant DNA reagent | pet21C-*Msm*RbpA | PMID: 28067618 | | |
| Recombinant DNA reagent | pet21C-*Mtb*RbpA | PMID: 28067618 | | |
| Recombinant DNA reagent | pET28a | EMD-Millipore (Novagen) | | |
| Recombinant DNA reagent | pET-SUMO *Msm*σA | PMID: 28067618 | | |
| Recombinant DNA reagent | pET-SUMO *Mtb*σA | PMID: 28067618 | | |
| Recombinant DNA reagent | pET-SUMO *Msm*RbpAΔNTT | PMID: 28067618 | | |
| Recombinant DNA reagent | pET-SUMO *Mtb*RbpAΔNTT | PMID: 28067618 | | |
| Recombinant DNA reagent | pGEMABC | PMID: 23389035 | Addgene 45398 | |
| Recombinant DNA reagent | pMP55 | this paper | | pAC22 derivative encoding Mtb RNAP with β S450Y. Derived from pAC22 by P69R substitution, removal of the N-terminal β insertion, and substitutions of increased predicted-strength RBSs for the rpoA, rpoZ, and rpoB::C RBSs. |
| Recombinant DNA reagent | pMP57 | this paper | | pMP55 with β Q1054H. Fdx resistant. |
| Recombinant DNA reagent | pMP61 | this paper | | pMP55 with wild-type S450 in place of β Y450. |
| Recombinant DNA reagent | pMP62 | this paper | | pMP61 with β S450L. |
| Recombinant DNA reagent | pRARE2 | EMD-Millipore (Novagen) | | |
| Recombinant DNA reagent | pUC57-AP3 | PMID: 25510492 | | |
| Chemical compound, drug | Fidaxomicin | VWR International, Inc. (Radnor, PA) | 2832–1 | |
| Chemical compound, drug | 3-([3-cholamidopropyl] dimethylammonio)−2-hydroxy-1-propanesulfonate (CHAPSO) | Sigma-Aldrich (St. Louis, MO) | C4695 | |
| Software, algorithm | Blocres | PMID: 23954653 | | |
| Software, algorithm | Chimera | PMID: 15264254 | | |
| Software, algorithm | Coot | PMID: 15572765 | | |
| Software, algorithm | CryoSPARC | PMID: 28165473 | | |

*Continued on next page*

*Continued*

| Reagent type (species) or resource | Designation | Source or reference | Identifiers | Additional information |
|---|---|---|---|---|
| Software, algorithm | EMAN2 | PMID: 16859925 | | |
| Software, algorithm | Gautomatch | http://www.mrc-lmb.cam.ac.uk/kzhang/Gautomatch | | |
| Software, algorithm | Gctf | PMID: 26592709 | | |
| Software, algorithm | Leginon | PMID: 20817100 | | |
| Software, algorithm | Molprobity | PMID: 20057044 | | |
| Software, algorithm | MotionCor2 | PMID: 28250466 | | |
| Software, algorithm | Phenix | PMID: 20124702 | | |
| Software, algorithm | PyMOL | Schrödinger, LLC (New York, NY) | http://www.pymol.org | |
| Software, algorithm | RELION | PMID: 23000701 | | |
| Software, algorithm | Serial EM | PMID: 16182563 | | |
| Software, algorithm | Unblur | PMID: 26023829 | | |
| Other | C-flat CF-1.2/1.3 400 mesh gold grids | Electron Microscopy Sciences (Hatfield, PA) | CF413-100-Au | |

## Protein expression and purification

*Mtb RNAP overexpression plasmid.* The overexpression plasmid (OEP) for *Mtb* RNAP was engineered from an existing OEP for *M. bovis* RNAP (*Czyz et al., 2014*), pAC22. Four modifications were made. First, a sequence in pAC22 that encodes an N-terminal 6-aa insertion at codon 2 of *rpoB* was removed. Second, a sequence upstream of *rpoZ* that included an ATG that potentially allowed an N-terminal extension on ω was removed. Third, to increase protein expression, the ribosome-binding sites (RBSs) for *rpoA*, *rpoZ*, and *rpoB::C* were re-engineered to encode stronger predicted RBSs using predicted translation initiation rates calculated using the Salis RBS strength calculator (https://salislab.net/software/) (*Espah Borujeni et al., 2014*). Finally, the single amino-acid difference between *Mtb* RNAP and *Mbo* RNAP at position 69 of β was changed from Pro (*Mbo*) to Arg (*Mtb*) (P69R). The resulting plasmid, pMP55, encodes β S450Y (Rif^R) *Mtb* RNAP. A wild-type derivative (Rif^S) was engineered by site-direct mutagenesis to give plasmid pMP61 that expresses the wild-type *Mtb* RNAP. A derivative of pMP55 encoding the β Q1054H substitution that confers resistance to Fidaxomicin (Fdx) (*Kurabachew et al., 2008*) was constructed by site-directed mutagenesis.

Samples for Cryo-EM grid preparation used *Mtb* His-tagged-$\sigma^A$ and RbpA co-expressed and purified as previously described (*Hubin et al., 2015*; *2017a*). To compare Fdx sensitivity of full-length RbpA and RbpA$^{\Delta NTT}$, these proteins, and *Mtb* His-tagged-$\sigma^A$ were expressed separately and purified as described previously (*Hubin et al., 2015*; *2017a*). Briefly, Rosetta-2 cells (EMD-Millipore/Novagen) were co-transformed with pET plasmids expressing *Mtb* $\sigma^A$(His-tagged) and RbpA and induced with 0.5 mM IPTG at 30°C for 4 hr. Clarified lysates was subjected to Ni$^{2+}$ affinity, removal of the His-tag, a second Ni$^{2+}$ affinity (collecting the flow through this time) and size exclusion chromatography.

*Mtb* RNAP was expressed and purified as previously described for *Mbo* and *Msm* RNAPs (*Davis et al., 2015*; *Hubin et al., 2017a*). *Eco* core RNAP, *Eco* $\sigma^{70}$, *Msm* $\sigma^A$, *Msm* RbpA, and *Msm* core RNAP were expressed and purified as described (*Davis et al., 2015*; *Hubin et al., 2015*; *2017a*).

## In vitro transcription assays

In vitro abortive initiation transcription assays were performed using the WT AP3 (−87 to +71) promoter at 37°C as described (*Davis et al., 2015*): Assays were performed in KCl assay buffer (10 mM Tris-HCl, pH 8.0, 50 mM KCl, 10 mM MgCl$_2$, 0.1 mM EDTA, 0.1 mM DTT, 50 μg-/mL BSA). The IC$_{50}$'s of Fdx on the different holos were calculated as follows: *Mtb* and *Msm* RNAP holo were

incubated with the cognate $\sigma^A$ and RbpA-FL or RbpA$^{\Delta NTT}$, and *Eco* RNAP (50 nM) was incubated with $\sigma^{70}$, to form holos. Holos were incubated with Fdx for 10 min at 37°C prior to addition of template DNA. DNA template was added (10 nM final) and the samples were incubated for 15 min at 37°C for open complex formation. Transcription was initiated with nucleotide mix, and stopped with a 2X Stop buffer (45 mM Tris-HCl, 45 mM Boric acid, 8 M Urea, 30 mM EDTA, 0.05% bromophenol blue, 0.05% xylene cyanol) after 10 min at 37°C. Transcription products were denatured by heating at 95°C for two minutes and visualized by polyacrylamide gel electrophoresis using phosphorimagery and quantified using ImageJ (*Schneider et al., 2012*).

## Agar disk diffusion assay

*Msm* strains MGM6232 (ΔrbpA attB::rbpA kan) and MGM6234 (ΔrbpA attB::rbpA(28-114) kan) (*Hubin et al., 2017a*) were grown overnight in LBsmeg (LB with 0.5% glycerol, 0.5% dextrose and 0.05% Tween$_{80}$) and 2 mL were centrifuged and resuspended in 200 µL of residual media and then plated. Filter discs were placed on the plates and stock solutions of Fdx were prepared in 10% DMSO at different concentrations (50 µM, 100 µM, 250 µM and 500 µM). 10 µl of antibiotic from each stock solution was pipetted onto the disks. Streptomycin (0.5 mg/ml, 860 µM) and 10% DMSO were used as positive and negative controls, respectively. Plates were incubated at 37°C for 74 hr and the zone of inhibition around each disk was photographed and measured.

## Preparation of Fdx/RbpA/$\sigma^A$-holo Complexes for Cryo-EM

*Mtb* RbpA/$\sigma^A$-holo (0.5 ml of 5 mg/ml) was injected into a Superose 6 Increase column (GE Healthcare Life Sciences, Pittsburgh, PA) equilibrated with 20 mM Tris-HCl pH 8.0, 150 mM K-Glutamate, 5 mM MgCl$_2$, 2.5 mM DTT. The peak fractions of the eluted protein were concentrated by centrifugal filtration (EMD-Millipore, Darmstadt, Germany) to 6 mg/mL protein concentration. Fdx (when used) was added at 100 µM and us-fork DNA (when used) was added to 20 µM. The samples were incubated on ice for 15 min and then 3-([3-cholamidopropyl]dimethylammonio)−2-hydroxy-1-propanesulfonate (CHAPSO) was added to the sample for a final concentration of 8 mM prior to grid preparation.

## Cryo-EM grid preparation

C-flat CF-1.2/1.3-4Au 400 mesh gold grids (Protochips, Morrisville, NC) were glow-discharged for 20 s prior to the application of 3.5 µl of the sample (4.0–6.0 mg/ml protein concentration). After blotting for 3–4.5 s, the grids were plunge-frozen in liquid ethane using an FEI Vitrobot Mark IV (FEI, Hillsboro, OR) with 100% chamber humidity at 22°C.

## Cryo-EM data acquisition and processing

Structural biology software was accessed through the SBGrid consortium (*Morin et al., 2013*).

*Fdx/RbpA/$\sigma^A$-holo/us-fork.* The grids were imaged using a 300 keV Titan Krios (FEI) equipped with a K2 Summit direct electron detector (Gatan, Warrendale, PA). Images were recorded with Leginon (*Nicholson et al., 2010*) in counting mode with a pixel size of 1.1 Å and a defocus range of 0.8 µm to 1.8 µm. Data were collected with a dose of 8 electrons/px/s. Images were recorded over a 10 s exposure with 0.2 s frames (50 total frames) to give a total dose of 66 electrons/Å$^2$. Dose-fractionated subframes were aligned and summed using MotionCor2 (*Zheng et al., 2017*) and subsequent dose-weighting was applied to each image. The contrast transfer function was estimated for each summed image using Gctf (*Zhang, 2016*). From the summed images, Gautomatch (developed by K. Zhang, MRC Laboratory of Molecular Biology, Cambridge, UK, http://www.mrc-lmb.cam.ac.uk/kzhang/Gautomatch) was used to pick particles with an auto-generated template. Autopicked particles were manually inspected, then subjected to 2D classification in cryoSPARC (*Punjani et al., 2017*) specifying 50 classes. Poorly populated and dimer classes were removed, resulting in a dataset of 582,169 particles. A subset of the dataset was used to generate an initial model of the complex in cryoSPARC (*ab-initio* reconstruction). Using the *ab-initio* model (low-pass filtered to 30 Å-resolution), particles were 3D classified into two classes using cryoSPARC heterogenous refinement. CryoSPARC homogenous refinement was performed for each class using the class map and corresponding particles, yielding two structures with different clamp conformations: open (Fdx/RbpA/$\sigma^A$-holo/us-fork; *Figure 1D*) and closed [RbpA/$\sigma^A$-holo/(us-fork)$_2$; *Figure 2A*]. Two rounds of

heterogenous/homogeneous refinements were performed for each class to achieve the highest resolution. The open class (Fdx/RbpA/σ$^A$-holo/us-fork) contained 173,509 particles with an overall resolution of 3.38 Å (*Figure 1—figure supplement 3A*) while the closed class [*Mtb* RNAP/σ$^A$/RbpA/(us-fork)$_2$] contained 171,547 paricles with a overall resolution of 3.27 Å (*Figure 2—figure supplement 1A*). Particle orientations of each class were plotted in cryoSPARC (*Figure 1—figure supplement 1C*, *Figure 2—figure supplement 1C*). FSC calculations (*Figure 1—figure supplement 1A*, *Figure 2—figure supplement 1A*) were performed in cryoSPARC and the half-map FSC (*Figure 1—figure supplement 1B*, *Figure 2—figure supplement 1B*) was calculated using EMAN2 (*Tang et al., 2007*). Local resolution calculations (*Figures 1E* and *2B*) were performed using blocres (*Cardone et al., 2013*).

Mtb *RbpA/σ$^A$-holo.* The grids were imaged using a 200 keV Talos Arctica (FEI) equipped with a K2 Summit direct electron detector (Gatan). Images were recorded with Serial EM (*Mastronarde, 2005*) in super-resolution counting mode with a super-resolution pixel size of 0.75 Å and a defocus range of 0.8 μm to 2.4 μm. Data were collected with a dose of 8 electrons/px/s. Images were recorded over a 15 s exposure using 0.3 s subframes (50 total frames) to give a total dose of 53 electrons/Å$^2$. Dose-fractionated subframes were 2 × 2 binned (giving a pixel size of 1.5 Å), aligned and summed using Unblur (*Grant and Grigorieff, 2015*). The contrast transfer function was estimated for each summed image using Gctf (*Zhang, 2016*). From the summed images, Gautomatch (developed by K. Zhang, MRC Laboratory of Molecular Biology, Cambridge, UK, http://www.mrc-lmb.cam.ac.uk/kzhang/Gautomatch) was used to pick particles with an auto-generated template. Autopicked particles were manually inspected, then subjected to 2D classification in RELION (*Scheres, 2012*) specifying 100 classes. Poorly populated classes were removed, resulting in a dataset of 289,154 particles. These particles were individually aligned across movie frames and dose-weighted using direct-detector-align_lmbfgs software to generate 'polished' particles (*Rubinstein and Brubaker, 2015*). A subset of the dataset was used to generate an initial model of the complex in cryoSPARC (*ab-initio* reconstruction). 'Polished' particles were 3D auto-refined in RELION using this *ab-initio* 3D template (low-pass filtered to 60 Å-resolution). RELION 3D classification into two classes was performed on the particles using the refined map and alignment angles. Among the 3D classes, the best-resolved class, containing 87,657 particles, was 3D auto-refined and post-processed in RELION. The overall resolution of this class was 6.9 Å (before post-processing) and 5.2 Å (after post-processing). Subsequent 3D classification did not improve resolution of this class.

*Fdx/RbpA/σ$^A$-holo.* The same procedure as described above for *Mtb* RbpA/σ$^A$-holo was used. After RELION 2D classification, poorly populated classes were removed, resulting in a dataset of 63,839 particles. In the end, the best-resolved 3D class, containing 21,115 particles, was 3D auto-refined and post-processed in RELION. The overall resolution of this class was 8.1 Å (before post-processing) and 6.5 Å (after post-processing).

## Model building and refinement

To build initial models of the protein components of the complex, *Msm* RbpA/σ$^A$-holo/us-fork structure (PDB ID 5TWI) (*Hubin et al., 2017a*) was manually fit into the cryo-EM density maps using Chimera (*Pettersen et al., 2004*) and real-space refined using Phenix (*Adams et al., 2010*). In the real-space refinement, domains of RNAP were rigid-body refined. For the high-resolution structures, the rigid-body refined models were subsequently refined with secondary structure restraints. A model of Fdx was generated from a crystal structure (*Serra et al., 2017*), edited in Phenix REEL, and refined into the cryo-EM density. Refined models were inspected and modified in Coot (*Emsley and Cowtan, 2004*) according to cryo-EM maps, followed by further real-space refinement with PHENIX.

## Acknowledgements

We thank K Uryu at The Rockefeller University Electron Microscopy Resource Center for help with EM sample preparation, M Ebrahim and J Sotiris at The Rockefeller University Cryo-EM Resource Center and LY Kim, M Kopylov, and E Eng at the New York Structural Biology Center for help with data collection, and members of our research groups for helpful comments on the manuscript. Some of the work presented here was conducted at the Simons Electron Microscopy Center and the National Resource for Automated Molecular Microscopy located at the New York Structural Biology

Center, supported by grants from the NIH National Institute of General Medical Sciences (GM103310), NYSTAR, and the Simons Foundation (349247). HB was supported by a Women and Science Postdoctoral Fellowship from The Rockefeller University. This work was supported by NIH grants R01 GM38660 to RL, R35 GM118130 to SAD, and R01 GM114450 to EAC.

## Additional information

### Funding

| Funder | Grant reference number | Author |
|---|---|---|
| Rockefeller University | Women in Science Fellowship | Hande Boyaci |
| National Institute of General Medical Sciences | R01 GM38660 | Robert Landick |
| National Institute of General Medical Sciences | R35 GM118130 | Seth A Darst |
| National Institute of General Medical Sciences | R01 GM114450 | Elizabeth A Campbell |

The funders had no role in study design, data collection and interpretation, or the decision to submit the work for publication.

### Author contributions

Hande Boyaci, James Chen, Mirjana Lilic, Margaret Palka, Rachel Anne Mooney, Investigation, Writing—review and editing; Robert Landick, Supervision, Funding acquisition, Writing—review and editing; Seth A Darst, Conceptualization, Supervision, Funding acquisition, Investigation, Writing—review and editing; Elizabeth A Campbell, Conceptualization, Supervision, Funding acquisition, Investigation, Writing—original draft, Writing—review and editing

### Author ORCIDs

Seth A Darst (iD) http://orcid.org/0000-0002-8241-3153

### Decision letter and Author response

Decision letter https://doi.org/10.7554/eLife.34823.029
Author response https://doi.org/10.7554/eLife.34823.030

## Additional files

### Supplementary files

• Supplementary file 1. Model statistics.
DOI: https://doi.org/10.7554/eLife.34823.013

• Supplementary file 2. Superimposition of cryo-EM structures.
DOI: https://doi.org/10.7554/eLife.34823.014

• Transparent reporting form
DOI: https://doi.org/10.7554/eLife.34823.015

### Major datasets

The following datasets were generated:

| Author(s) | Year | Dataset title | Dataset URL | Database, license, and accessibility information |
|---|---|---|---|---|
| Boyaci H, Chen J, Lilic M, Darst SA, Campbell EA | 2018 | Mycobacterium tuberculosis RNAP Holo/RbpA/Fidaxomicin | http://www.rcsb.org/pdb/search/structid-Search.do?structureId=6C06 | Publicly available at the RCSB Protein Data Bank (accession no: 6C06) |

| Boyaci H, Chen J, Lilic M, Darst SA, Campbell EA | 2018 | Mycobacterium tuberculosis RNAP Holo/RbpA in relaxed state | http://www.rcsb.org/pdb/search/structid-Search.do?structureId=6C05 | Publicly available at the RCSB Protein Data Bank (accession no: 6C05) |
| Boyaci H, Chen J, Lilic M, Darst SA, Campbell EA | 2018 | Mycobacterium tuberculosis RNAP Holo/RbpA/us-fork DNA | http://www.rcsb.org/pdb/search/structid-Search.do?structureId=6C04 | Publicly available at the RCSB Protein Data Bank (accession no: 6C04) |
| Boyaci H, Chen J, Lilic M, Darst SA, Campbell EA | 2018 | Mycobacterium tuberculosis Fdx/RNAP Holo/RbpA/us-fork DNA | http://www.rcsb.org/pdb/search/structid-Search.do?structureId=6BZO | Publicly available at the RCSB Protein Data Bank (accession no: 6BZO) |

The following previously published datasets were used:

| Author(s) | Year | Dataset title | Dataset URL | Database, license, and accessibility information |
|---|---|---|---|---|
| Hubin EA, Darst SA, Campbell EA | 2017 | Crystal structure of a Mycobacterium smegmatis transcription initiation complex with RbpA | http://www.rcsb.org/pdb/explore/explore.do?structureId=5tw1 | Publicly available at the RCSB Protein Data Bank (accession no: 5TW1) |
| Darst SA, Campbell EA, Lilic M | 2017 | Structure of Mycobacterium smegmatis transcription initiation complex with a full transcription bubble | https://www.rcsb.org/structure/5vi5 | Publicly available at the RCSB Protein Data Bank (accession no: 5VI5) |

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
