## [Decision Letter]

Thank you for submitting your article "Fidaxomicin jams *M. tuberculosis* RNA polymerase motions needed for initiation via RbpA contacts" for consideration by *eLife*. Your article has been reviewed by three peer reviewers, and the evaluation has been overseen by Gisela Storz as the Reviewing and Senior Editor. The following individuals involved in review of your submission have agreed to reveal their identity: William R Jacobs (Reviewer #1); Yuan He (Reviewer #2); Dong Wang (Reviewer #3).

The reviewers have discussed the reviews with one another and the Reviewing Editor has drafted this decision to help you prepare a revised submission.

Summary:

A critical challenge for tuberculosis (TB) treatment and other antimicrobial clinical practice is to develop novel antibiotics that can outcompete increasing drug resistance. Inhibition of bacteria RNA polymerase (RNAP) transcription and subsequent bacteria growth is a proven effective therapeutic strategy. Fidaxomicin (Fdx) is a narrow spectrum antibiotic that target Gram-positive anaerobes and mycobacteria (including Mtb) RNA polymerase (RNAP).

In this work, Campbell and colleagues build on previous studies reporting the first CryoEM structures of a complete *M. tuberculosis* RNAP holoenzyme in complex with Fdx at 3.4 A resolution. In fact, the authors reported a total of four cryoEM structures: Two fdx bound Mtb /RbpA/σA-holo complexes (with or without us-fork promoter DNA) and two fdx absence RbpA/σA -holo complexes (with or without us-fork promoter DNA). By comparing the different RNAP conformational states from four solution complexes of the same RNAP, authors are able to obtain several important structural insights into mechanism of action of Fdx.

Overall the reviewers were in agreement that this paper summarizes an impressive amount of work and is easy to read.

Essential revisions:

The reviewers had a limited number of suggestions that should be addressed in a revised version.

1) Expand the discussion to suggest how understanding of the structure could apply to improving the poor bioactivity and the use of this drug to treat *M. tuberculosis* infections.

2) The paper by Tupin et al., 2010, presented a nice and detailed mechanistic investigation of the inhibition of bacterial RNA polymerase by Lpm (Fdx). This is study should be cited and discussed.

3) The section titled "The RbpA^NTT^ is critical for Fdx potency against Mtb RNAP in vitro and in vivo" is misleading since M. smegmatis was used for the growth inhibition assay. While M. smegmatis is a model organism to study *M. tuberculosis*, it is not *M. tuberculosis*. Suggest using *M. tuberculosis* safe strains to test the growth inhibition of *M. tuberculosis* by Fdx or avoid this direct implication.

---

## [Author Response]

Essential revisions:

The reviewers had a limited number of suggestions that should be addressed in a revised version.1) Expand the discussion to suggest how understanding of the structure could apply to improving the poor bioactivity and the use of this drug to treat M. tuberculosis infections.

The structural results define the details of a binding determinant in the RNAP that facilitates the discovery of other inhibitors (including Fdx derivatives) that have activity and might have better bioavailability. Because we now know which atoms of Fdx interact with the protein, one can now make educated guesses on which chemical groups to modify. It is beyond the scope of our expertise to say what changes can be made but we propose the structure can now be used to guide modification or screens. We have made some edits to change the ‘tone’ of our manuscript with this in mind. Changes have been made to the Abstract, Introduction, and Summary section.

2) The paper by Tupin et al., EMBO J (2010) 29, 2527-2537, presented a nice and detailed mechanistic investigation of the inhibition of bacterial RNA polymerase by Lpm (Fdx). This is study should be cited and discussed.

We thank the reviewers for pointing out this oversight. We have added a sentence pointing out how the mechanistic analyses of Fdx action from the Brodolin group (actually in two papers, Tupin et al., 2010 and Morichaud et al., 2016) are broadly consistent with our conclusions (subsection “Fdx inhibits RNAP by trapping an open-clamp conformation” last sentence).

3) The section titled "The RbpANTT is critical for Fdx potency against Mtb RNAP in vitro and in vivo" is misleading since M. smegmatis was used for the growth inhibition assay. While M. smegmatis is a model organism to study M. tuberculosis, it is not M. tuberculosis. Suggest using M. tuberculosis safe strains to test the growth inhibition of M. tuberculosis by Fdx or avoid this direct implication.

This was an oversight; we thank the reviewers for catching it. The section title has been changed to ‘The RbpA^NTT^ is critical for Fdx potency against mycobacterial RNAP in vitroand in vivo’.